# Intracranial Aneurysms: Relevance of Superposed Blood Pulse Waves and Tobacco Smoke?

Ulrich Barz [1],*, Almut Schreiber [2] and Helmut Barz [3]

[1] Clinic for Internal Medicine, Helios Medical Center Pirna, Struppener Str. 13, D-01796 Pirna, Germany
[2] Medical Center for Occupational and Social Hygiene, Fit for Work, Bautzener Landstraße 91, D-01324 Dresden, Germany
[3] Department of Neuropathology, Dietrich Bonhoeffer Hospital, D-17022 Neubrandenburg, Germany
* Correspondence: ulrichbarz@yahoo.de

**Abstract:** Background: Intracranial aneurysms (IAs) are found in around 3–4% of elderly people. The authors attempt to answer why IAs develop exclusively in the circle of Willis (CW) and why IAs in the frontal cerebral arteries are unusually frequent in men. Methods: The location and frequency of IAs were analyzed using relevant publications (MEDLINE and PubMed). Results: It is suggested that superposed blood pulse waves may have an influence on the development of IAs. The superposition of blood pulse waves is caused by the meeting of the bilateral cerebral arteries in the CW. The predominance of IAs in women is striking (about 1.7:1). However, IAs in the anterior cerebral arteries and anterior communicating artery are significantly more common in men than in women (approximately 1.8:1). The authors hypothesize that greater nicotine abuse in men may explain this phenomenon. Cigarette smoke apparently reaches the anterior cerebral arteries via the olfactory pathway. Conclusion: It seems possible that superposed pulse waves are a substantial factor in the occurrence of IAs. The toxic effects of tobacco smoke appear to have greater impact on IA development than the sex-specific influences that are responsible for the predominance of IAs in women.

**Keywords:** intracranial aneurysm; circle of Willis; superposed blood pulse waves; nicotine abuse; subarachnoid hemorrhage

## 1. Introduction

Intracranial (saccular or berry) aneurysms (IAs) are often found incidentally when performing cerebral angiographies. They are the most common cause of subarachnoid hemorrhage (SAH), with a lethality rate of up to 50% [1]. The most important factors in IA development are high blood pressure and nicotine abuse [2–6]. It has also been argued that aneurysms develop due to wall inflammation or locally increased hemodynamic wall shear stress at the arterial branches or bifurcations, leading to degeneration of the internal elastic membrane (IEM) [1,5]. High blood pressure values or high wall shear stress can cause an acute or prolonged rupture of the IEM in the arterial wall [6–8]. In studies of mice and rats, cerebral aneurysms can be induced by renal blood hypertension and unilateral ligation of a carotid artery. In these experiments, the rupture of the IEM is the trigger of aneurysm development [9–11].

It Is noteworthy that atherosclerosis is not a major pathogenic factor in the development of IAs. Severe atherosclerosis leads to distended and highly tortuous arteries. Although dolichoectasia most commonly affects the basilar artery and the vertebral arteries, it can also be found in other cerebral arteries. Cerebral dolichoectasia is often referred to as fusiform or dilatative arteriopathy [12,13].

Regardless of the fact that aneurysms occur predominantly in women [5,14], two questions arise in the present study:

Why do IAs occur almost exclusively in and near the circle of Willis (CW)?

Why do IAs in the anterior cerebral arteries (ACAs) primarily affect men?

To answer these two questions, two things are considered: the hemodynamic influence of superposed blood pulse waves in the arteries of the CW and the influence of tobacco abuse on the brain arteries.

The meeting of opposing blood pulse waves in the CW leads to a particularly strong local wall shear stress and can thus be a factor in IA formation. Tobacco smoke is absorbed not only in the lungs and oral mucosa, but also in the mucosa of the paranasal sinuses. Hypothetically, the toxic gases can diffuse into the leptomeninges via the olfactory tract. This may explain why IAs are much more frequent in the ACAs than in other cerebral arteries [2,14].

## 2. Materials and Methods

Publications on IAs and SAH were evaluated using the search tools MEDLINE and PubMed. Special attention was paid to papers or reviews with large patient numbers and detailed information on the location of aneurysms. In a total of 21,995 subjects (12,136 men and 9859 women), 916 IAs were found (540/5.5% of women and 376/3.1% of men) (see Table 2). A total of 2162 people with SAH were observed (1362/63% women and 800/37% men) [3,15]. Publications that were not included in the evaluation were those with small case numbers and those without information on aneurysm location. Aneurysms in vascular provinces outside the brain and arteriosclerotic aneurysms were not analyzed. The statistical *p*-values listed in this paper are taken from the cited publications.

## 3. Results

### 3.1. Prevalence and Rupture Risk of IAs

In a total of 94,912 people from 21 countries, an overall prevalence of non-perforated IAs was found in 3.2% of people [16].

IAs grow slowly. The number and size of aneurysms increases with age [17]. In a study of consecutive Magnetic resonance angiographies in 3414 people, almost half of the IAs (47.9%) had a diameter of less than 3 mm [18]. In a series of 8696 Magnetic resonance angiographies in healthy adults, IAs of 2 mm or more were detected in 3.2% of subjects. The prevalence in people under 50 years of age was 2.7% in women and 1.9% in men. In those aged over 50, the prevalence was 5.4% in women and 2.8% in men [19]. In a study of 4813 adults aged 35–75 years from Shanghai, China, a three-dimensional time-of-flight Magnetic resonance angiography detected non-ruptured IAs in 5.5% of men and 8.4% of women [20].

The average incidence of SAH in an international study was about 10 per 100,000 people (with the lowest incidence rate of 2 per 100,000 people in China and the highest incidence rate of 22.5 per 100,000 people in Finland) [21].

From these data, it can be concluded that (regardless of the size of the aneurysm) about one in 300–400 IAs ruptures.

### 3.2. Superposed Blood Pulse Waves Can Promote the Development of Aneurysms

The anatomical image of the CW, simplified, is an arterial ring formed by the four major cerebral arteries at the base of the brain (Figure 1).

Occlusion of one of the four cerebral arteries (e.g., due to an embolism) can be compensated for by the other three arteries. However, the CW allows for the phenomenon of opposing blood pressure waves (superposed pulse waves), which is uncommon in the blood supply to other organs. The unique feature of the CW is that the communicating arteries connect the bilateral or anterior and posterior cerebral arteries in a tubular fashion, allowing the pulse waves to add up [22] (Figure 2).

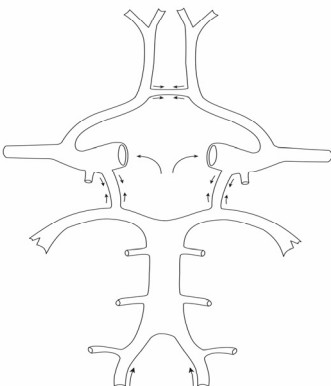

**Figure 1.** Circle of Willis. The cerebral arteries of both sides (internal carotid arteries and vertebral arteries) are connected by communicating arteries.

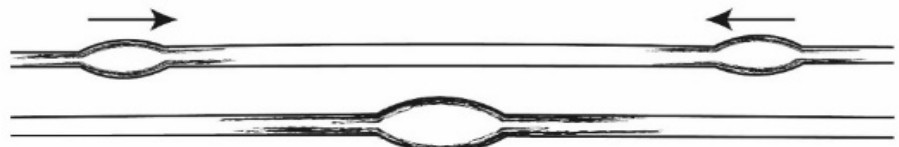

**Figure 2.** Superposed pulse wave. Schematic picture of two opposing pulse waves.

To our knowledge, this patho-dynamic detail of superposed pulse waves in the development of aneurysms has not been previously discussed. Opposing waves intensify pressure on the vessel walls, regardless of whether or not liquid is flowing in the tube vessels. (Bernoulli's equation; the static pressure on the (blood) vessel wall increases the lower the velocity of the fluid flow in the tubes). Depending on the elastic properties of the arteries, the speed of the pulse waves can be up to 10 times faster than the blood flow in the arteries [22]. Similar to a rubber membrane, the IEM is stretched and reflects the pulse waves that are triggered by cardiac activity. In contrast to the easily deformable tunica media of the arterial walls (smooth muscle), the IEM is primarily responsible for balancing the wall tension caused by blood pressure and can tear under a heavy load. Figure 3 shows a ruptured IA with a fatal SAH.

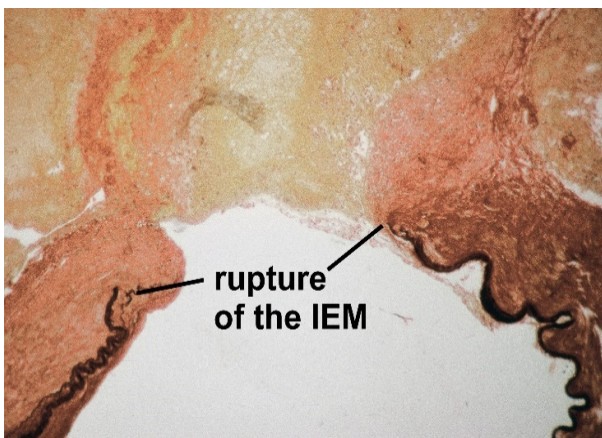

**Figure 3.** Rupture of the internal elastic membrane (IEM) and rupture of the (saccular) aneurysm of the middle cerebral artery. Fatal subarachnoid hemorrhage in an elderly man. (Elastica van Gieson staining, ×20).

The authors argue that vascular branching is more likely to lead to atherosclerotic changes due to the possible turbulence of blood flow.

### 3.3. Different Frequency of IAs on the Left and Right Side of the Brain Base

If superposed pulse waves foster the occurrence of IAs, it is very likely that most aneurysms will be found at or near the midline of the CW and near the communicating arteries. In a series of 1073 ruptured aneurysms in China, 690 (64.3%) were found in close proximity to the anterior and posterior communicating arteries [15]. In another study of ruptured aneurysms in the Hong Kong Chinese population, 49% of aneurysms (130 of 266) were also found near the anterior and posterior communicating arteries [23].

Unfortunately, most publications lack information on which side of the brain base the aneurysms were located on. Only four publications were found that specified whether the aneurysms were located on the left or right side of the CW (Table 1).

**Table 1.** Side differences of aneurysms in the circle of Willis.

|  | *n* | **Artery** | **Left Side** | **Right Side** | *p*-**Value** |
|---|---|---|---|---|---|
| Aarhus [24] | 17 | vertebrobasilar | 15 | 2 | <0.0001 |
| Liu et al. [25] | 69 | anterior cerebral | 53 | 16 | <0.001 |
| Krzyzewski [26] | 99 | internal carotid | 64 | 35 | <0.01 |
| Imaizumi [27] | 148 | internal carotid | 82 | 66 | |

The statistically confirmed dominance of left-sided aneurysms in three papers is surprising. No convincing explanation for this phenomenon could be found in the four publications that specified the side that the aneurysms were on. Aarhus et al. (2009) suggest that the left-sided *dominant* vertebral artery or of the vertebral arteries may be responsible for the higher incidence of ruptured left-asymmetry sided aneurysms in the vertebral arteries [24]. The authors of the present paper have reservations about the attempt to explain the development of aneurysms by different diameters of the arteries. It is well-known that anomalies of the CW are the rule rather than the exception.

### 3.4. Possible Explanation for the Predominance of Left-Sided IAs

We assume that there is an anatomical explanation for the predominance of left-sided aneurysms at the base of the brain. The blood pulse waves from the left-sided cerebral arteries take slightly longer to reach the CW than those from the right-sided cerebral arteries. This is because the left-sided arteries leading to the brain leave the aortic arch a few centimeters after the (right-sided) brachiocephalic trunk (Figure 4).

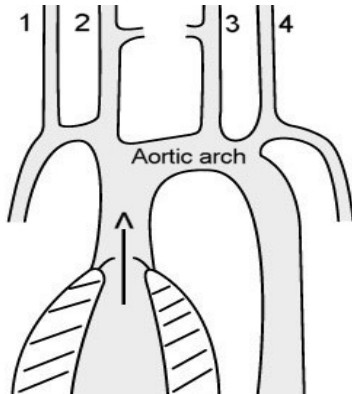

**Figure 4.** Aortic arch. The four major arteries of the brain are as follows: the right vertebral artery (1), the right internal carotid artery (2), the left internal carotid artery (3), and the left vertebral artery (4). The pulse waves from the left-sided brain arteries reach the circle of Willis at a slightly later time than the pulse waves from the right-sided brain arteries.

This difference in the path of the cerebral blood pulse waves can cause the pulse waves of each side to meet more frequently on the left side of the CW than the right. The more frequent detection of IAs on the left side of the CW supports the hypothesis that superposed blood pulse waves may be a factor in the development of IAs.

*3.5. Frequency of IAs in Women and Men*

IAs are significantly more prevalent in women than men (Table 2).

**Table 2.** Frequency of intracranial aneurysms (IA) in women and men.

| | Women *n* | Men *n* | Women IA | Men IA | IA Women % | IA Men % |
|---|---|---|---|---|---|---|
| Harada [19] | 3253 | 5341 | 144 | 133 | 4.4 | 2.5 |
| Imaizumi [27] | 1707 | 2363 | 106 | 70 | 6.2 | 3.0 |
| Li [20] | 2445 | 2368 | 206 | 130 | 8.6 | 5.5 |
| Horikoshi [28] | 2454 | 2064 | 84 | 43 | 3.4 | 2.1 |
| Total | 9859 | 12,136 | 540 | 376 | 5.3 | 3.1 |
| Ratio | | | | | 1.7 | 1 |

Answering why IAs are more prevalent in women (Table 2) would go beyond the scope of this paper. The prevalence of IAs in women is addressed surprisingly little in the literature and has not been satisfactorily explained [3]. The authors suggest that tissue edema may play an important role in the development of arterial disease during both the menstrual cycle and pregnancy.

Patients with SAH also shows a similar distribution between women and men: of 1256 patients with SAH, 784 (62.4%) were women and 472 (37.6%) were men (1.7:1) [15]. Another publication showed the same ratio in 906 cases of SAH: 578 (63.8%) occurred in women and 328 (36.2%) in men (1.8:1) [3].

*3.6. Unusually High Prevalence of IAs in Anterior Cerebral Arteries in Men*

Although IAs are much more common in women, IAs in the ACAs are significantly more common in men. Aarhus et al. (2009) write: "It is our impression that anterior communicating artery aneurysms present more frequently as the cause of an acute SAH compared with aneurysms in other locations." [24] (Table 3).

**Table 3.** Aneurysms in the frontal arteries.

| | *n* (IA) ACA + AcoA | Men % | Women % | *p*-Value |
|---|---|---|---|---|
| Kongable [3] | 301 | 46.1 | 26.6 | <0.001 |
| Ghods [14] | 97 | 29.0 | 15.0 | <0.001 |
| Hamdan [29] | 216 | 44.8 | 26.3 | <0.001 |
| Aarhus [24] | 130 | 47.7 | 25.7 | <0.0001 |
| Krzyzewski [26] | 79 | 31.0 | 17.0 | <0.015 |
| Harada [19] | 57 | 18.0 | 8.9 | <0.001 |

Predominance of aneurysms in men (mean ratio 1.8:1). IA: intracranial aneurysms, ACA: anterior cerebral artery, and AcoA: anterior communicating artery.

However, it remains unclear why IAs in the ACAs are particularly prevalent in men. An explanation for this striking anomaly could not be found in the scientific literature.

*3.7. Tobacco Smoke: A Toxic Factor at the Anterior Cerebral Arteries?*

An unknown local factor is most likely responsible for the increased fragility of the ACAs in men. A possible explanation could be the more pronounced nicotine abuse in men, even though women are smoking more and more frequently. In a Polish study of 357 patients with IAs, 33.2% of 232 women and 53.6% of 125 men were smokers. Women smoked 15 and men smoked 21 cigarettes per day [26]. Nicotine abuse is a known pathogenic factor in the development of IAs [2,3,6,26]. Chronic exposure to cigarette smoke causes damage to the endothelium of blood vessels [30]. Tobacco abuse causes irreversible damage to vessel walls, and the cessation of nicotine use after years does not significantly reduce the likelihood of wall rupture [31]. We hypothesize that tobacco smoke enters the frontal leptomeninges by diffusion. Possible pathways include the mucosa of the paranasal sinuses (ethmoidal cells) and along the olfactory system through the lamina cribrosa. Consequently, the constituents of tobacco smoke might contact the ACAs rather than other arteries of the brain. This hypothesis is also supported by the fact that the sense of smell is significantly impaired in smokers [32]. A surprising finding illustrates the potential toxic effect of tobacco smoke on the ACAs. In the age group of 60 years and older, the mean diameter of ruptured aneurysms in the ACAs was significantly smaller (5.06 mm, $n = 77$) than the mean diameter of ruptured aneurysms in all other arteries at the base of the brain (7.29 mm, $n = 138$; $p < 0.001$). "Therefore, aneurysms located in the anterior communicating artery and the distal ACA may bleed before they reach a larger size . . . " [33]. Ruptured aneurysms in the anterior communicating artery and ACAs were significantly smaller than those of the posterior and middle cerebral arteries ($p < 0.01$) [34]. It is very likely that the smaller diameter of the ruptured IAs indicates more regressive changes in the walls of the ACAs and the IAs formed there. Accordingly, the mean age of patients with ruptured IAs ($n = 361$; men 170, women 191) is significantly lower in men than in women (48.6 years vs. 53.8 years; $p < 0.005$) [24].

Studies show that malignant brain gliomas are more common in smokers than in non-smokers [35–37] and that they are significantly more common in men than in women (1.6:1) [38]. Men showed a higher incidence of glioblastoma in the frontal lobe than in the temporal lobe (57.7% vs. 47.3%) whereas women had a higher incidence of glioblastomas in the temporal lobe than in the frontal lobe (52.7% vs. 42.3%, $p = 0.024$) [39].

Consistent treatment of arterial hypertension and a decrease in nicotine abuse have significantly reduced the incidence of IAs in recent decades. In a study, which involved more than 8000 people, the number of aneurysmal SAHs decreased by about 40% between 1980 and 2010 [6].

## 4. Conclusions

The authors present two hypotheses:

1. The superposition of bilateral blood pulse waves may be a reason for the occurrence of IAs in the CW. The increased incidence of aneurysms on the left side of the CW supports this hypothesis.

2. The statistically proven prevalence of aneurysms in the ACAs in men may be explained by nicotine abuse.

## 5. Limitations of the Study

The prevalence of aneurysms on the left side of the CW (Table 1) is an assumption based on only four publications (three of which are statistically validated). Further studies are needed to substantiate these data.

Further studies are also needed to validate the hypothesis that tobacco smoke may be a cause of the unusually high incidence of aneurysms in men in the ACAs.

## 6. Clinical Impact

1. This paper discusses the little-known phenomenon of the superposition of blood pulse waves in the CW. It seems possible that superposed pulse waves are a substantial factor in the development of IAs at the base of the brain.

2. Tobacco abuse not only carries the risk of carcinoma formation in various organs, but is likely a factor in the development of aneurysms at the base of the brain as well. The local effect of tobacco smoke seems to have a greater influence on the development of aneurysms rather than the sex-specific influences that are responsible for the predominance of IAs in women.

**Author Contributions:** Conceptualization: U.B.; curation of data: U.B., H.B. and A.S.; formal analysis: H.B.; methodology: U.B., A.S. and H.B.; project management: U.B.; visualization: U.B., A.S. and H.B.; writing—original draft: U.B.; writing—revision and editing: U.B., A.S. and H.B. All authors have read and agreed to the published version of the manuscript.

**Funding:** This research received no external funding.

**Institutional Review Board Statement:** The article describes a study with human subjects. The article does not include information about living or identifiable individuals. Ethical approval was not needed for analysis of retrospective anonymous data.

**Informed Consent Statement:** Not applicable.

**Data Availability Statement:** Data sharing not applicable. No new data were created or analyzed in this study.

**Acknowledgments:** The authors thank Kristen McLean for her help with translating the paper from German to English.

**Conflicts of Interest:** The authors declare that they have no conflict of interest.

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
