# Peer review of "Intracranial Aneurysms: Relevance of Superposed Blood Pulse Waves and Tobacco Smoke?"

_2813-2475, doi:10.3390/jvd2020016_

Round 1

Reviewer 1 Report (New Reviewer)

The paper under review is concerned with a possible answer to the question: why IAs (intracranial aneurysms) develop exclusively in the circle of Willis and why IAs in the frontal cerebral arteries occur unusually but frequently in men. The content of the paper is interesting in my opinion. The following are my comments/suggestions.

In the abstract, I suggest using italic/bold option for writing "Background", "Methods", "Results", "Conclusion". Or these words should be started with a new line except the first one, that is "Background". 

In-text citation style should be unified, either author-date style or numerical citation style but not both; for example see the sections: "Introduction" and  "Results".

Improve the quality of Figure 1.

In Table 1, what about the p-value regarding the last row?

I think the text "Table 3." in the first row of Table 3 is extra.

Author Response

The authors (UB AS HB) thank the reviewers (Reviewer 1 and 2) for their evaluations.

Response to report of Reviewer 1 

  1. minor spelling corrections needed: Answer:The paper was sent to a colleague (English-speaking colleague from Canada) for correction. Any changes or improvements were incorporated.
  2. Does the introduction provide enough background information and does it include all relevant references? Answer: The two questions raised in lines 51 and 52 have not yet been formulated or answered in the literature. These are matters of great pathophysiological interest. No references were found for these questions.
  3. Are all cited references relevant to the research? Answer: Yes, only publications useful for explaining or justifying the two theses or questions were included in the reference list.
  4. Italic/bold option for the abstract: Answer: methods, results and conclusion have been changed to italic style: Background, Methods, Results, Conclusion.

5) The citation style in the text should be unified: Answer: with exceptions (Aarhus...) the text has been changed to numerical citation style. 

6) Improve the quality of Figure 1 ANSWER:... New  figure

7) What about the p-value in the last row in Table 1? Answer:  The difference between 82 (left hand side) and 62 (right hand side) is obviously not significant. A p-value determination was not given or not carried out in the paper by Imaizumi et al..

In Table 3: Tab 3 was removed from the first row of Table 

Reviewer 2 Report (New Reviewer)

Intracranial aneurysms (IAs) are often found in the circle of Willis (CW), the most important factors in IA development are high blood pressure and nicotine abuse. The authors attempted to answer why IAs develop exclusively in the CW and why IAs in the frontal cerebral arteries occur unusually frequent in men. And they had interesting results: the superposition of blood pulse waves were a substantial factor in the occurrence of IAs, and the nicotine might have greater impact on ACAs in men. The conclusion brings the reader novelty information about the IAs, but there were still some questions.

1.     The viewpoint of superposition of pulse waves was speculated by the clinical data from literatures, the weakness of the argument was that there was no provable direct link;

2.     The authors hypothesized that tobacco smoke enters the frontal leptomeninges by diffusion. The hypothesis might be one of the reasons for brain tumors, such as meningiomas or gliomas, but might be lower correlation to the ACAs, the internal elastic membrane was damaged by the nicotine from the frontal leptomeninges?  This assumption might be invalid, more proof was needed to support the hypothesis.

Author Response

Response to report of reviewer 2

  1. Is the research design appropriate and are the methods adequately described? Answer:

The work is based on two facts, both of which are known, but which have been little or not discussed so far. The authors present two hypotheses:

1a: Aneurysms occur almost exclusively at the base of the brain in arteries, that appear to be healthy (at least without significant arteriosclerotic changes). A haemodynamic cause is plausible. Experimental data on this is not available and cannot be performed by the authors. The hypothetical explanation of an increased stress on the inner elastic membrane due to a superposition of pulse waves seems possible.

1b: It is difficult to prove that cigarette smoke can be a cause of aneurysms in the brain. The statistical data at the anterior cerebral arteries suggest that it is. But the authors wanted to explain why the occurrence of aneurysms in the anterior cerebral arteries is diametrically different from that in the other cerebral arteries. Overall, aneurysms in the cerebral arteries (including the frontal arteries!) occur much more frequently in women than in men (1.8:1), while the distribution is reversed in the frontal arteries (men to women 1.7:1). As long as no other cause is found, the smoking habits of both sexes could be decisive. The fact that women are smoking more and more frequently will probably (unfortunately) offset the differences somewhat in the future. However, the occurrence of aneurysms will continue to increase overall.

  1. Are the conclusions supported by the results? Answer: The two hypotheses are supported by the frequency and location of the aneurysms.
  2. The weakness of the argument (superposition of pulse waves) was that there was no provable direct link.: Answer: The authors have only drawn attention to the addition of pressure waves with counter-propagating pulse waves. Physically, the intravascular static pressure on the vessel wall increases even with greatly reduced flow velocity (Bernoulli's equatione).
  3. The tabacco smoke hypothesis might be lower correlation to the ACAs internal elastic membrane was damaged by the nicotine from the frontal leptomeninges? Answer: Of course, this is also a hypothesis. The cerebral arteries literally bathe in the cerebrospinal fluid in the subarachnoid space. The question of diffusion of tobacco smoke is best answered by experimental studies.

Round 2

Reviewer 2 Report (New Reviewer)

The prevalence of tabacco smoke was higher in men than women, and the nicotine abuse caused damage to the endothelium of blood vessels, the hypothesis might be that the aneurysm would be higher in men than in women in all positions, so I did not agree that the hypothesis in this manuscript without direct evidence.

This manuscript is a resubmission of an earlier submission. The following is a list of the peer review reports and author responses from that submission.

Round 1

Reviewer 1 Report

All reported hypothesis are simple assumptions without any scientific support. This paper appears, as it is, written by people without a deep knowledge of what they are reporting.

Author Response

Answers to your reply, please see the pdf:  "jvd-2076332-answers to reviewer"  

Reviewer 2 Report

1. In methods: the number of searched publications, included vs. excluded should be declared. 

2.Describe how could you conclude that one in 300-400 IAs rupture from the given data.

3. Figure 1: abbreviations in the figure should be reported in the figure's legend.

4. In conclusion: the prevalence of male predominance of ACA aneurysms due to tobacco smoking should be considered in the conclusion as a hypothesis, just like the first conclusion, needing to be supported by evidence. 

Author Response

Reply to Rewiewer 2

REWIEWER 2: Comments to and Suggestions for Authors:
1. In methods: the number of searched publications, included vs. excluded should be
declared.
2.Describe how could you conclude that one in 300-400 IAs rupture from the given data.
3. Figure 1: abbreviations in the figure should be reported in the figure's legend.
4. In conclusion: the prevalence of male predominance of ACA aneurysms due to tobacco
smoking should be considered in the conclusion as a hypothesis, just like the first
conclusion, needing to be supported by evidence

Reply (red: changed in the paper)

  1. In methods: the number of searched publications, included vs. excluded should be
    declared. (Line 57)

Our study was not designed to include as many publications as possible, but to find a sufficient number of publications relevant to the topics addressed in the paper. These publications were included. These were publications with large numbers of cases and statistically valid results. Aneurysms caused by arteriosclerotic changes (aorta, iliac arteries) were not analyzed. No publications on superposed pulse waves as a pathogenic factor in the development of intracranial (berry) aneurysms were found.

It is well known that smoking habits are more prevalent in men than in women. That women might catch up in the near future is possible but not (yet) relevant to the publication.  The essential point about the sex distribution was that IAs in the anterior cerebral arteries dominated in men. That a similar effect can be assumed in female smokers is by no means excluded.

  1. Describe how could you conclude that one in 300-400 IAs rupture from the given data.

(Line 75)

Vlak et al. (2011) describe an overall prevalence of 3.2% for unruptured intracranial IAs in a total of 94,912 people from 21 countries [16]. Harada et al. (2013) found IAs with a size of 2 mm or greater in 3.2% of subjects in a series of 8,696 MR angiographies in healthy adults [19]. The prevalence in people younger than 50 was 2.7% in women and 1.9% in men. In those over 50, the prevalence was 5.4% in women and 2.8% in men [19]. Li et al. (2013) described a prevalence of unruptured IAs of 5.5% in men and 8.4% in women (7.0% overall) in 4813 adults aged 35 to 75 years using three-dimensional time-of-flight magnetic resonance angiography [27]. This study suggests that unruptured aneurysms are probably even more common in China than described by Vlak et al. or Harada et al. [16,19].

Ingall et al. (2000) found that the average incidence of SAH in an international study was about 10 per 100,000 people (with the lowest incidence rate of 2 per 100,000 people in China and the highest incidence rate of 22.5 per 100,000 people in Finland) [20].

From these data, we conclude that (regardless of aneurysm size) approximately one in 300-400 IAs ruptures.  Not in the paper: If 3.2% of 100,000 people have an aneurysm (3,200 people) and if about 10 of 100 000 people experience SAH, the rupture frequency of IAs is 1:320.  

  1. Figure 1: abbreviations in the figure should be reported in the figure's legend.

(LINE 94)

The abbreviations in Figure 1 are reproduced in the legend as follows: A. com. Ant.; anterior communicating artery; Aa. car. int.: internal carotid arteries;  Aa. com. post.: posterior communicating arteries; A. bas.: basilar artery; Aa. verteb.: vertebral arteries.

Figure 1. was used in an earlier seminar by Dr. H. Barz and included in the publication. 

  1. In Conclusion: The prevalence of male predominance of ACA aneurysms due to tobacco
    smoking should be considered in the conclusion as a hypothesis, just like the first
    conclusion, needing to be supported by evidence.

We complete: (Line 224)

The presence of aneurysms at the base of the brain supports the hypothesis that superposed blood pulse waves could be a cause of IA formation. The increased occurrence of IAs on the left side of the CW supports this hypothesis.

Hypothetically, the statistically proven prevalence of ACA aneurysms in men can be explained by higher nicotine abuse in men.

  1. Limitations of the study (Line 230)

The prevalence of aneurysms on the left side of the CW (table 1) is an assumption based on only four publications (3 of which are statistically validated). Further studies are needed to substantiate this data.

The hypothesis that the unusually frequent occurrence of aneurysms in men in the anterior cerebral arteries (including the anterior communicating artery) is attributed to tobacco smoke, should be supported by further studies.
